# “Beyond Safer Injecting”—Health and Social Needs and Acceptance of Support among Clients of a Supervised Injecting Facility

**DOI:** 10.3390/ijerph16112032

**Published:** 2019-06-07

**Authors:** Vendula Belackova, Edmund Silins, Allison M. Salmon, Marianne Jauncey, Carolyn A. Day

**Affiliations:** 1Uniting Medically Supervised Injecting Centre, 66 Darlinghurst Road, Potts Point, NSW 2011, Australia; vendulabelackova@gmail.com (V.B.); allisonsalmon@gmail.com (A.M.S.); mjauncey@uniting.org (M.J.); carolyn.day@sydney.edu.au (C.A.D.); 2National Drug and Alcohol Research Centre, University of New South Wales, High St, Kensington, NSW 2052, Australia; 3Addiction Medicine, Sydney Central Clinical School, University of Sydney, Sydney, NSW 2006, Australia

**Keywords:** supervised injecting facility, high-risk drug use, people who inject drugs, health and social needs, support services, harm reduction

## Abstract

Health and social issues in aging populations of people who inject drugs (PWID) tend to aggregate, despite risky injecting practices decreasing with age. Identifying needs and avenues of support is becoming increasingly important. We described the health and social situation among clients of a long-running supervised injecting facility (SIF) in Sydney, Australia. An interviewer-administered survey (*n* = 182) assessed current housing status, employment, physical and mental health, incarceration history, drug use, engagement in drug treatment, health service utilization, and willingness to accept support. Results were compared to the information provided at initial visit. Up to half of the participants transitioned between lower- and higher-risk health and social indicators over time. Willingness to accept support was greatest amongst those with higher self-perceived need. Support for mental health was a low priority, despite the high self-reporting of mental health issues. SIF clients are open to support for health and social issues, despite ongoing active drug use. Lower-threshold services such as SIFs are well-positioned to recognize and respond to deteriorating health and social issues for PWID. Facilitating care and treatment remains a challenge when the services to which people are being referred are higher-threshold with a more rigid approach.

## 1. Introduction

Harm reduction programs are evidence-based interventions which reduce both individual and societal harms of drug use and change risk environments [1,2,3]. Supervised injecting facilities (SIFs) and Drug Consumption Rooms (DCRs) are harm reduction programs which provide space for safer drug administration in hygienic settings under the supervision of qualified staff [4], and are increasing in number worldwide [5]. In 2016, 92 SIFs/DCRs operated across 11 countries, with the majority in Germany, the Netherlands and Switzerland, several in Spain, Denmark, Norway and France, and, until recently, one each in Australia, Canada and Luxembourg [5]. A ‘trial’ SIF/DCR opened in Melbourne, Australia in July 2018 [5], and the Canadian Ministry of Health has thus far approved 33 new Supervised Consumption Services between 2016 and 2018 with further applications being assessed [5]. The opening of SIFs in Portugal is scheduled for 2018/19 and is under discussion in Ireland, Scotland and the US [5].

SIF clients have been described as regular injectors who are, when they first attend the service, at a higher risk of infectious diseases, illegal sources of income, incarceration, unsafe injecting practices and unstable accommodation than infrequent- or non-attenders [6,7,8]. Accordingly, the aims and activities of SIFs commonly span beyond safer drug administration, and include referrals into drug treatment and health and social services [9,10,11,12]. 

Uniting’s Sydney Medically Supervised Injecting Centre (referred to as Sydney SIF herein) is the first and longest-running supervised injecting facility in the Southern Hemisphere, and over 16,000 clients have registered with the service since 2001. Sydney SIF clients manifest high levels of marginalization and unstable housing [13]. Overall, there are indications that the population of people who inject drugs (PWID) is aging in Australia [14,15] and elsewhere [16]. While risky injecting practices decrease with age [17,18], health and social issues in aging populations of PWID tend to aggregate [16,19,20]. Furthermore, the marginalization of PWID impacts on the intent to seek treatment as well as treatment retention [16]. This means that identifying SIF clients’ needs and avenues for support is becoming increasingly important.

The Sydney SIF, like other SIFs and drug consumption rooms (DCRs), operates on harm reduction principles and ‘meets clients where they are at’. The SIF does not impose any mandatory health and social interventions on clients beyond the opportunity to use drugs in a safer and more hygienic manner. Referrals to health and social services, including drug treatment, at the Sydney SIF are similar to other low-threshold programs, in that they are opportunistic, require the identification of clients’ needs and can benefit from trusting relationships between staff and clients. Research has indicated that the clients who attend the Sydney SIF more frequently are more likely to accept a referral to health and social services [11,13], but these investigations have been limited in scope and have not directly addressed the changes, if any, in clients’ needs since their initial visit. In this study, we aimed to assess: (1) the current health and social needs of Sydney SIF clients; (2) transitions between lower- and higher-risk health and social indicators since the initial visit; and, (3) the determinants of seeking support. 

## 2. Materials and Methods

The Sydney SIF clinical model has been described previously [21]. Briefly, clients enter the reception area (Stage 1) and must register the first time they use the service. Clients do not provide identification, but a unique identifier is assigned to them. At all subsequent visits, clients declare their unique identifier and the drug they intend to inject and are assessed against admission criteria (aged ≥18 years, not pregnant, have a history of injecting drugs, and are not intoxicated). Clients then proceed to Stage 2 where they inject substances that they procured offsite under medical supervision in booths; all injecting equipment is provided. Following injection, clients move to the aftercare area (Stage 3) and continue to be monitored by clinical staff until they leave the premises. The service operates seven days per week and is open from 09:30 to 21:30 on weekdays and from 09:30 to 17:30 on weekends. Frontline staff are comprised of registered nurses and health education officers.

The detailed health and social information collected from clients at registration during their initial visit is not routinely updated. A 42-item questionnaire to update that information was developed, which was based on the questions asked at registration. The interviewer-administered questionnaire assessed a range of indicators including: housing status (current accommodation type), employment and source of income (work status, income source including welfare, crime and sex work), physical (hepatitis C status and testing) and mental health (seen a doctor, psychiatrist or counsellor for any mental health issues), incarceration history, drug use in the last four weeks (mode of administration and frequency of injecting), overdose history, engagement in drug treatment (12 treatment modality types, with an option for other), and health service utilization of 15 local services spanning primary healthcare, social support, mental health, drug treatment, PWID peer organizations, hospital emergency departments, with an option of other in the last 12 months. 

Individuals who reported an issue in relation to any of the indicators assessed were asked if they would like support (e.g., ‘Would you like to talk to one of our staff about this issue?’; Response categories: ‘Yes, now if possible’, ‘Yes, on my next visit’, ‘No’, and ‘Don’t know’). Participants were also asked what was the most important issue for them at the moment, and whether they would like to talk to one of the staff about it (response categories as above).

The questionnaire was reviewed by the Sydney SIF Consumer Action Group (CAG), a peer-based special interest group which regularly provides important feedback on project-related work being undertaken at the SIF. Members reviewed the interview questionnaires and schedule and provided feedback on the language, appropriateness and topic areas, and reviewed the study findings.

Clients in Stage 3 were approached by a SIF staff member who informed them about the study and determined eligibility. People were eligible to participate if they had attended the service during the data collection period (October–November 2017), had attended the service on at least one other day, and had not already participated. Participation was voluntary. The questionnaire was administered by a SIF staff member in a private clinic room. Interviews took approximately 20 min to complete. Answers were confidential. In line with established practice, participants were remunerated AU$20. Interviews were conducted across all opening hours and on 19 days during the data collection period. All individuals gave their informed consent for inclusion before they participated in the study. Clients who were unable to provide informed consent due to their level of intoxication were approached again later or on another day. The study was conducted in accordance with the Declaration of Helsinki, and the protocol was reviewed by the Human Research Ethics Committee of the South Eastern Sydney Local Health District (17/207).

There were three parts to the analysis. Firstly, we compared aggregate health and social indicators reported at the time of the survey to the aggregate health and social indicators reported previously at initial visit, using the t-test for the equality of means. Secondly, we investigated individual transitions between lower- and higher-risk health and social indicators at initial visit and at the time of the survey. For each health and social issue and service utilization item, individuals were categorized into those who reported the issue: (1) only at initial visit; (2) only at the time of the survey; (3) at both time-points; and, (4) at neither time-point. The proportions of individuals in each category were reported. Finally, we performed a factor analysis across all relevant variables from the survey and service records to identify the factors which were associated with individuals’ willingness to discuss with staff: (1) housing issues; (2) drug use and treatment; and, (3) take-home naloxone. Variables from the factor analysis that yielded factor loadings of ≥0.25 were included in separate stepwise logistic regression models to identify the characteristics which were significantly associated (*p* < 0.1) with willingness to discuss each of the three issues with staff. Stata 14 was used for all analyses.

## 3. Results

### 3.1. Sample

The demographic, drug use and service utilization characteristics of survey participants (*n* = 182) are shown in Table 1. Survey participants were mostly male (69%), aged 43 years on average (SD = 9.0) and had been injecting drugs since age 19 years (SD = 6.9). The drugs that the participants used most frequently at the Sydney SIF were heroin (54%) and methamphetamine (31%). Seventeen percent of participants self-identified as Aboriginal and/or Torres Strait Islander. Participants had, on average, been clients of the service for nine years. The mean number of visits since their initial visit was about 800. In the past 12 months, participants had received an average of two referrals to health and social services (Table 1).

We interviewed 36% (*n* = 182) of all clients (*n* = 500) who visited the Sydney SIF during the study period. Survey participants and non-participants were remarkably similar in characteristics. There were no differences between participants and non-participants in terms of gender, age, Aboriginality, drug used most often, and age at first injection. However, survey participants were longer and more frequent users of the service, and were more likely to report a health issue at their initial visit than non-participants (Table 1).

### 3.2. Aggregate Health and Social Indicators

The aggregate health and social indicators among survey participants at initial visit and at the time of the survey are shown in Table 2. Unstable housing was common among survey participants at both time-points (about 40%). However, there were marked differences between time-points in other health and social indicators. A comparison showed that the proportion of participants unemployed (92% versus 80%; *p* < 0.01), on government income support (88% versus 73%; *p* < 0.001), and experiencing a physical (69% versus 38%; *p* < 0.01) or mental health issue (64% versus 18%; *p* < 0.001) was notably higher at the time of the survey than previously at initial visit. The proportion of participants attending a nearby local community health service (73% versus 33%; *p* < 0.01) and engaging in drug treatment (93% versus 61%; *p* < 0.01) also increased between the two time-points (Table 2). 

### 3.3. Transitions between Lower- and Higher-Risk Health and Social Indicators

Individual transitions between lower- and higher-risk health and social indicators at initial visit and at the time of the survey are shown in Figure 1. Between initial visit and the time of the survey, there were transitions in all health and social indicators, the largest being in relation to health and housing issues. For example, up to one-third (35%) of participants had a health issue, and up to one-quarter (23%) of participants were in unstable housing at one time-point only (Figure 1).

In comparison, there was less fluctuation in terms of employment and the receipt of government income support, as substantial proportions of participants reported unemployment (76%) and were receiving government income support (68%) at both time-points. Subsequently, the proportion of participants transitioning between lower- and higher-risk indicators within those domains were smaller than for health and housing (up to 20% of participants were receiving government income support and up to 16% of participants reported unemployment at one time-point only). In terms of drug use, more than half (56%) of participants reported injecting opioids at both time-points. While thirty-eight percent of individuals reported daily injecting at both time-points, up to one-quarter (24%) of participants were daily drug injectors at one time-point only. About half (48%) of participants reported that for the first time since their initial visit to the Sydney SIF they had attended a nearby primary healthcare service which targeted services to more marginalized communities, including PWID (Figure 1).

### 3.4. Current Issues and Extent of Support Requested

The most important current issue and the extent of support requested for that issue among survey participants is shown in Figure 2. The most important current issues were housing (26%), family or social issues (19%) and drug treatment or reducing drug use (18%). However, of participants who voiced an important current issue (*n* = 158), not all requested support for that issue (e.g., 26% reported housing as an important issue and only 13% requested housing support). Eight percent of participants reported that they were not experiencing any major issue at the moment (Figure 2). For each category, there was no significant difference between the proportion of males and females indicating that they were experiencing the issue.

Participants were also asked if they were interested to receive support for six key health and social issues that the Sydney SIF is well-positioned to influence (Table 3). About two-thirds (63%) of participants requested support in at least one of the six key areas. Overall, the most common request for support related to take-home naloxone training (48%). Just under half (44%) of participants who had not previously received training were interested in being trained. The next most common request for support was for hepatitis C testing and treatment (25%). Most participants had previously been tested for hepatitis C. Among the minority that had not been tested, 18% were interested in testing and treatment (Table 3). Support for mental health (12%) was the least requested (Table 3), despite the high self-report of current mental health issues (64%, Table 2).

### 3.5. Factors Associated with Willingness to Discuss Issues with Staff

The factors associated with willingness to discuss housing issues, drug use and treatment, and take-home naloxone training with staff are reported in Table 4. Individuals who expressed a willingness to talk to staff about housing were those who were currently in unstable accommodation (Odds Ratio (OR) 5.57, *p* = 0.009) and had expressed housing as being a main issue (OR 12.74, *p* < 0.001). Individuals who were willing to discuss drug use and treatment with staff were more likely to be predominantly heroin users (OR 3.90, *p* = 0.020) and had expressed drug use and treatment as being their main issue (OR 3.18, *p* = 0.026), and were less likely to be currently in treatment (OR 0.43, *p* = 0.044). The two factors significantly associated with willingness to discuss take-home naloxone training were being of Aboriginal and/or Torres Strait Islander origin (OR 4.41, *p* = 0.024) and having experienced an overdose elsewhere other than at the Sydney SIF (OR 5.12, *p* = 0.008) (Table 4).

## 4. Discussion

This study found that, over time, clients’ engagement with treatment and a local low-threshold primary healthcare service increased, a direct or indirect outcome of referrals they received at the Sydney SIF. The health and social situation remained stable for a large proportion of clients and improved for others. Nevertheless, over time, the overall health and social situation among people who attended the Sydney SIF deteriorated, highlighting the dynamic and transitional nature of clients’ needs over the course of their SIF involvement. This is perhaps not surprising given clients’ ongoing (although at times intermittent) substance use which is a highly stigmatized and criminalized behavior, and likely to impact on their health situation, hamper employment opportunities and increase contact with police [22].

The large increase over time in the number of people who were engaged in drug treatment (from 61% at initial visit to 93% at the time of the survey) challenges the misconception that SIFs encourage drug use and do not do enough to link people into treatment [23]. On the other hand, if people who are engaged in treatment are still using the SIF then is the treatment working? Critics of SIFs often point to a false dichotomy between treatment and harm reduction [24], and believe both cannot exist simultaneously. In reality, SIFs can do both [25]. Treatment does not have to be the primary goal and support is growing for targeted low-threshold healthcare and treatment which offers services without attempting to control drug use [26].

The risk of overdose increases with the length of time injecting [27], so it is unsurprising that more participants had experienced a drug overdose (at the SIF or elsewhere) at the time of the survey than at their initial visit to the SIF. However, the onsite immediate overdose management at the SIF [21,28] may well have contributed to individuals staying alive longer over their, on average, 20-year (predominantly opioid) injecting histories [5].

While a proportion (about two-thirds) of people were open to discussing support options for managing their health and social issues, a minority who voiced a current issue were uninterested in support at the time. This is perhaps not surprising given clients’ primary purpose for presenting to the SIF is to administer their drugs. For some, however, this may be because the issue of concern is being addressed by other services (e.g., an accommodation support service is regularly co-located onsite at the Sydney SIF). It also highlights the inability of low-threshold services to enforce health and social service referrals onto unwilling clients [2,29,30].

Individuals more open to support were those who were homeless, used heroin, had experienced overdose, and self-identified need. Not surprisingly, unstable accommodation and experience of overdose outside the SIF were strongly associated with willingness to discuss housing issues and take-home naloxone training, respectively. It is unclear why people who are Aboriginal or Torres Strait Islander were more open to support, but it suggests that this marginalized population are willing to engage and warrant attention. Research from different international settings has consistently shown that more marginalized people use SIFs [5]. Despite different settings [5], frequent SIF attendance increased the likelihood of referral to health and social services [11,13] and engagement in drug treatment [31,32]. However, we found that such indicators of SIF service use were weaker determinants for requesting support than variables indicative of marginalization (i.e., homelessness, heroin use), self-identified needs and existing contact with relevant services.

The challenge of asking about and addressing peoples’ identified needs in low-threshold settings is how to do so without inadvertently alienating people by making them feel they are problems to be solved. A tenet of low-threshold harm reduction services which service people who are highly marginalized is the ability to meet people ‘where they are at’. This means providing a low barrier to accessing care and support. It also means working with people in a flexible and consumer-focused way and accepting consumers as adequate and complete as they are, and that they do not need to change anything to be accepted and valid individuals. That can be hard to do, but it is essential to effective engagement and gaining trust. It is also an essential pre-condition to help people who are stigmatized to build confidence and motivation to make adaptive changes, even changes that they themselves have identified as wanting. The first step needs to be authentic humanistic engagement. The risk of attempting to be proactive in providing support and problem solving before invitations to do so is that it inadvertently reinforces stigma and creates additional barriers to care for the consumer. 

The nature of the support sought by clients may also challenge the service model. Canadian overdose prevention sites are a response to the opioid overdose crisis and focus on providing wide-spread safer injecting spaces; they may lack the capacity to support health and social issues of clients [33]. Globally, SIFs also exist as stand-alone facilities within the immediate vicinity of supportive services, for example, the Sydney SIF is located near a low-threshold primary healthcare service. Other examples include co-locating SIFs at detoxification services (i.e., Vancouver Insite [34]) and healthcare facilities [35]. The Melbourne SIF is now based within a community health service on a public housing estate.

However, distance from support services is unlikely to be the main obstacle to care. Drug treatment and hospital services are often high-threshold programs which are constrained by abstinence-orientated approaches, inflexibility, and top-down organized programs which stigmatize people who use drugs [2,36]. In Australia, consumers have expressed discontent, even with methadone maintenance programs for their lack of client-centered approaches [37]. Transitioning from a low-threshold service might be a large step for SIF clients and referral efforts from harm reduction programs can be hampered by the practices of high-threshold programs. This issue warrants more careful consideration in terms of how referrals from SIFs and other harm reduction services are managed and the way services which aim to address the needs of this population are developed. 

There are several limitations to this study. The surveyed sample represented about one-third of all clients who attended the SIF during the study period. Participants were clients with longer histories of SIF use, a greater number of visits, and accordingly, more referrals, which might have reduced the variability in the data and the significance of predictors in the logit models. The resulting sample size was small when including all relevant variables and their respective missing values. The potential for social desirability response bias may have influenced results. Similarly, data may reflect the health and social situation of higher-functioning SIF clients, as only people who attended the service were interviewed. The overall deterioration in client situation as well as their increased access to services was of note; a more rigorous study design (e.g., a cohort study) could assess determinants of those individual trajectories. Also, several questions in the survey were phrased differently than the questionnaire used at initial visit, limiting the strength of some comparisons. For example, for some items there were differences in the assessment periods (e.g., past week versus past month) and response categories (e.g., frequency categories of drug use, and types of health services accessed) between questions asked at initial visit (which for some individuals were based on items developed more than 10 years ago) and questions asked at the time of the survey.

## 5. Conclusions

SIF clients connect with health and social services over time, and the referrals they receive onsite likely contribute to this. Many SIF clients are open to health and social support at SIFs, even though they present primarily to administer drugs in a safer setting. However, the overall health and social situation of SIF clients is likely to deteriorate over time, albeit not uniformly. Clients who are most likely to respond to support are those with greatest need. Nevertheless, engaging and referring clients from low-threshold programs is just the first step in linking them to appropriate care. More attention is needed to develop improved practices at the high-threshold programs to which they are referred if these opportunities are to be harnessed.

## Figures and Tables

**Figure 1 ijerph-16-02032-f001:**
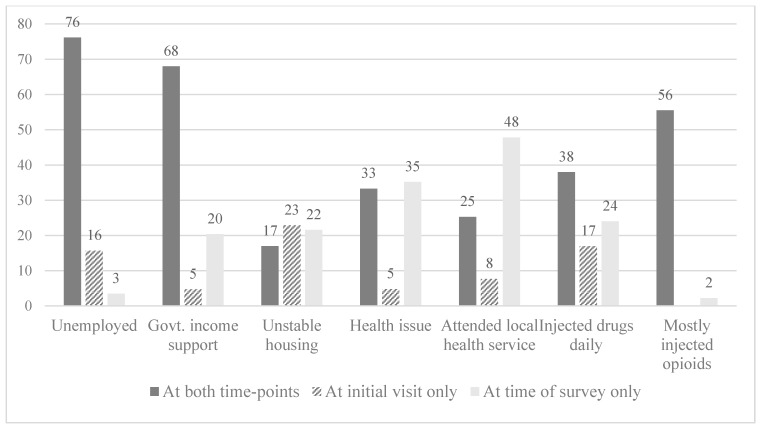
Individual transitions between lower- and higher-risk health and social indicators at initial visit and at the time of the survey (%).

**Figure 2 ijerph-16-02032-f002:**
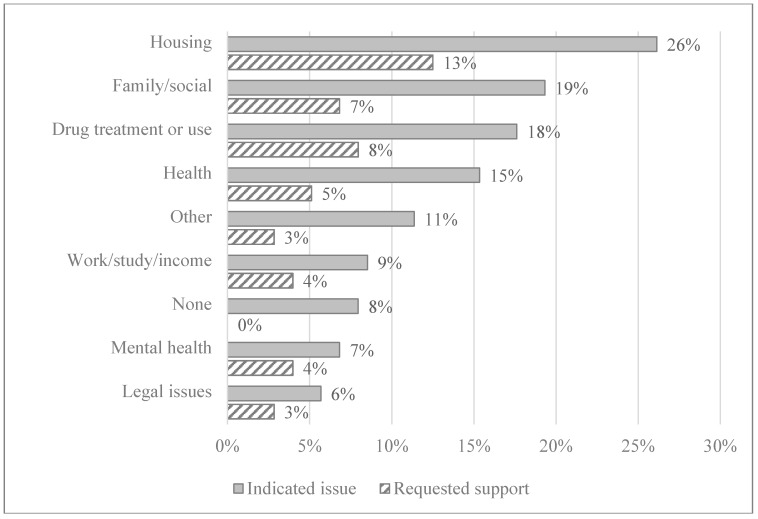
Most important current issues and the extent of support requested among survey participants (%).

**Table 1 ijerph-16-02032-t001:** Demographic, drug use and service utilization characteristics of survey participants and non-participants.

Characteristic	Survey Participants *n* = 182 % (*n*/*N*)	Non-Participants *n* = 318 % (*n*/*N*)
Gender		
Female	29 (53)	24 (77)
Male	69 (126)	74 (235)
Transgender	1 (3)	2 (5)
Not stated	0 (0)	<1 (2)
Age, in years—median (mean; SD)	42 (43.0; 9.0)	43 (42.6; 6.9)
Aboriginality		
Neither Aboriginal nor Torres Strait Islander	78 (141)	73 (232)
Aboriginal and/or Torres Strait Islander	17 (31)	16 (50)
Did not specify	2 (3)	1 (3)
Missing	4 (7)	10 (31)
Other characteristics (assessed at initial visit)		
Stable accommodation	60 (92/153)	67 (86/172)
Employed	20 (35/174)	22 (63/288)
Ever in drug treatment	61 (111/182)	59 (188/318)
Ever in prison	45 (77/94)	41 (117/284)
Ever overdosed	38 (64/169)	39 (108/278)
Health issue	36 (52/143)	29 (67/229)
Injected drugs daily *	55 (94/171)	45 (124/277)
Client of local primary health care service **	33 (60/182)	23 (74/318)
Drug used most (assessed in the 8 months before survey) ^a^		
Heroin	54	52
Oxycodone	5	5
Buprenorphine **	8	3
Methadone	6	8
Morphine	6	6
Cocaine	4	2
Methamphetamine	31	40
Service use—median (mean; SD)		
Number of visits since initial visit ***	313 (796.2; 1308.6)	97 (406.2; 811.6)
Number of visits during survey period ***	5 (9.0; 10.4)	2 (2.9; 3.9)
Number of referrals in the past 12 months ***	1 (2.0; 2.7)	0 (0.6; 1.4)
Number of years since initial visit **	10.5 (9.4; 8.5)	8.3 (8.2; 7.6)
Age in years when first injected	18 (19.3; 6.9)	18 (19.6; 7.0)

^a^ Among drugs listed; * *p* < 0.05; ** *p* < 0.01; *** *p* < 0.001.

**Table 2 ijerph-16-02032-t002:** Aggregate health and social indicators among survey participants at the time of the survey and at initial visit.

Health and Social Indicators	*N*	At the Time of the Survey %	At Initial Visit %	*t*-Test
Unstable housing	153	39	40	−0.242
Unemployed	172	92	80	−3.795 **
Government income support	172	88	73	0.000 ***
Currently has a physical health issue	105	69	38	−5.608 **
Currently has a mental health issue	176	64	18	0.000 ***
Attendance at a nearby primary healthcare service	182	73	33	−8.596 **
Injected daily in the past month	171	62	55	−1.438
Injected mostly opioid	182	58	56	−2.017 *
Engaged in drug treatment	181	93	61	−8.778 **
Has been to prison	171	73	45	−9.936 **
Has had an overdose	167	61	38	5.187 **

* *p* < 0.05; ** *p* < 0.01; *** *p* < 0.001.

**Table 3 ijerph-16-02032-t003:** Proportion of survey participants interested to receive support for key health and social issues that the Sydney SIF is well-positioned to influence.

	Interested in Support Now or at Next Visit	Not Interested in Support or Does Not Know
	% (*n*/*N*)	% (*n*/*N*)
Take-home naloxone training		
All participants	48 (51/106)	52 (55/106)
Participants not previously trained	44 (41/93)	56 (52/93)
Hepatitis C testing and treatment		
All participants	25 (41/166)	75 (125/166)
Participants not previously tested	18 (2/11)	82 (9/11)
Drug treatment	21 (38/181)	79 (143/181)
Physical health support	21 (29/135)	79 (106/135)
Accommodation	20 (36/182)	80 (146/182)
Mental health support	12 (19/165)	88 (146/165)
Support (%)		
Any	63	37
Excluding take-home naloxone training	52	48

**Table 4 ijerph-16-02032-t004:** Factors associated with willingness to discuss housing issues, drug use and treatment, and take-home naloxone training with staff (Adjusted OR, *p* value).

	Willingness to Discuss with Staff AOR (*p* Value)		
	Housing Issues	Drug Use and Treatment	Take-Home Naloxone Training
Aboriginal and/or Torres Strait Islander origin ^1^	-	-	**4.41 (0.024)**
Heroin used most at the Sydney SIF	-	**3.90 (0.02) ^1^**	3.62 (0.054) ^2^
Methamphetamine used most the Sydney SIF	-	2.27 (0.086)	3.46 (0.069)
Buprenorphine used most at the Sydney SIF	**0.068 (0.042)**	-	-
Oxycodone used in past month at the Sydney SIF	3.25 (0.068)	-	-
Expressed this issue as the most important	**12.74 (p < 0.001) ^3^**	**3.18 (0.026) ^4^**	-
Currently in unstable accommodation	**5.57 (0.009)**	-	-
Currently in drug treatment	-	**0.43 (0.044)**	-
Ever in drug treatment	-	0.42 (0.087)	-
Previously received take-home naloxone training	-	2.28 (0.057)	-
Experienced overdose outside of the Sydney SIF	3.32 (0.072)	-	**5.12 (0.008)**
Experienced overdose for the first time since initial visit	-	-	0.34 (0.081)
* Number of observations*	*n = 112*	*n = 159*	*n = 78 ^5^*
* LR chi^2^, df, (p)*	*38.33, 5, (p < 0.001)*	*25.65, 6, (0.003)*	*15.47, 5, (0.0085)*
* Pseudo R^2^*	*0.345*	*0.1530*	*0.1431*

^1^ Self-report, ^2^ Database records, ^3^ Housing, ^4^ Reducing drug use and/or access to treatment, ^5^ Only included those who were not trained already. Note: multivariate logistic regression models used with stepwise selection (*p* < 0.1); Adjusted odds ratios (AOR); Supervised Injecting Facility (SIF); Likelihood ratio (LR); Degrees of freedom (df).

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
