# Peer review of "“Beyond Safer Injecting”—Health and Social Needs and Acceptance of Support among Clients of a Supervised Injecting Facility"

_ijerph, 2019, doi:10.3390/ijerph16112032_

Round 1

Reviewer 1 Report

RE: ijerph-505264

Thank you for the opportunity to review this interesting paper on the health and social needs of people using the Sydney SIF.  The data will be of interest not only to people here in Australia where SIFs have doubled in size (from 1 to 2) in the last 12 months but also a number of other settings globally. 

The introductory two paragraphs could be framed with a bit more context about the diversity of the available SIFs/DRCs internationally. 

Some evidence on the ageing population of people who inject drugs would also be helpful in para 3 – the AIHW cite includes very few PWID so hard to know that it shows an ageing cohort.  There is also some more recent evidence on the health and social issues of ageing that could be included – this review may help to identify some Rosen, D., et al. (2011). "Characteristics and consequences of heroin use among older adults in the United States: A review of the literature, treatment implications, and recommendations for further research." Addictive Behaviors 36(4): 279-285.

Methods: Some narrative of the approach to participation may be helpful – it is noted that the response rate is included in the narrative of the results section as well as in table 1.  This may well be important to include in the discussion as those responding were using the SIF an average 3 times more than the people who did not respond.  The non-participants also appear less likely to have had a referral made in the previous 12 months. 

Results; Any reason why the data were not analysed by gender/sex? Even a mention that there were no differences if that is the case is important.  

‘On social benefits’ may be better described as ‘government income support’ rather than ‘welafre benefits’ as in the narrative.   

For table 2 the *** is missing for p<0.001.

Section 3.3: is the nearby community health service Kirkteon rd Centre?  if so this may be better described as a  primary health care service targeting their services to more marginalised communities including PWID.  This is also noted in table 2 ‘attendance at a local community health service’

Table 3: it is not clear how the 80/176 who answered the TH naloxone question is reported.  May be worth considering a line in the tables that includes the 80/176 already done training.  You could also consider the same for the hepatitis c testing and treatment question as no doubt many people will have already done treatment or know their current negative status because of previous testing.

Is there any data on the proportion of overdoses that took place at the SIF – of so it may also be worth including as it is noted in para 2 of the discussion that SIF management of overdose ‘may have contributed to individuals staying alive longer’. 

Discussion: I wonder if it is worth framing this on the improvement and remained stable ‘large proportion’ rather than on the people for which their health and social situation had ‘deteriorated’. I understand from your discussion and conclusion that you do not want to add further to the stigma and barriers to access that already exist for many using the Sydney SIF.

The one thing in table two that stick out for me that wasn’t really discussed was the huge increase in the number of people who were ‘engaged in treatment’ – from 61% to 93%.  It may be that there is a different understanding of what treatment means for people over the years they have been using the service. This is an important finding and is worth highlighting more if it is possible to understand the nuances of ‘engaged in treatment’.  Partly this is because it is one of the criticisms of SIFs that they do not work hard enough to get people in to treatment – but alternatively it may be problematic in that almost all the people are ‘engaged in treatment’ but they are still using the SIF regularly so the treatment is not working.   The need to describe the benefits of access to low threshold ‘treatment’ may be required here. 

Para 4 (lines 225-231),  How was ‘at risk’ defined? Hy would heroin use necessarily be a risk? Nothing on concomitant use of benzos and or alcohol which may be considered ‘a risk’.  The ‘previous research’ mentioned mixes Canadian data and Australian data despite the differences in drug market settings – this may be sorted by some more detailed discussion of the diversity of SIFs suggested for the introduction. 

Psara 6:  it may also be worth mentioning that the Richmond SIF now is based within a ‘community health service’ on the North Richmond public housing estate.  I did try and find a cite – perhaps Jauncey, M. 2017. If not us, Who? Drug and Alcohol Review 36(S1):5.  

Limitations: It may also be worth highlighting the potential for social desirability bias as people want to show the team that they really do want to ‘make adaptive changes’.  Also it may be that people who are not functioning as well (much much worse) than they were 10 years ago are now no longer able to even get to the SIF so these people have not been included in the data. 

Refences:

Note that the cite 5 has now been published in DAR as well.

Author Response

Response to Reviewer 1 comments

Thank you for the opportunity to review this interesting paper on the health and social needs of people using the Sydney SIF.  The data will be of interest not only to people here in Australia where SIFs have doubled in size (from 1 to 2) in the last 12 months but also a number of other settings globally. 

Point 1: The introductory two paragraphs could be framed with a bit more context about the diversity of the available SIFs/DRCs internationally. 

Response 1: The first introductory paragraph now includes the international context re SIFs/DCRs, “In 2016, 92 SIFs/DCRs operated across 11 countries, with the majority in Germany, Netherlands and Switzerland, several in Spain, Denmark, Norway and France and, until recently, one each in Australia, Canada and Luxembourg [5]. A ‘trial’ SIF/DCR opened in Melbourne, Australia in July 2018 [5] and the Canadian Ministry of Health has thus far approved 33 new Supervised Consumption Services between 2016 and 2018 with further applications being assessed {Belackova, 2019 #34}. The opening of SIFs in Portugal is scheduled for 2018/19 and is under discussion in Ireland, Scotland and the US [5].”

Point 2: Some evidence on the ageing population of people who inject drugs would also be helpful in para 3 – the AIHW cite includes very few PWID so hard to know that it shows an ageing cohort.  There is also some more recent evidence on the health and social issues of ageing that could be included – this review may help to identify some Rosen, D., et al. (2011). "Characteristics and consequences of heroin use among older adults in the United States: A review of the literature, treatment implications, and recommendations for further research." Addictive Behaviors 36(4): 279-285.

Response 2: We thank the reviewer for drawing attention to this review, the findings (now cited) add to the Introduction, “Furthermore, the marginalization of PWID impacts on the intent to seek treatment as well as treatment retention {Rosen, 2011 #35}.” We have also provided more relevant and up to date references to support indications that the population of people who inject drugs is aging in Australia:

Topp, L., Day, C. A., Iversen, J., Wand, H., & Maher, L. (2011). Fifteen years of HIV surveillance among people who inject drugs: The Australian Needle and Syringe Program Survey 1995-2009 AIDS, 25, 835-842.

Larney, S., Hickman, M., Guy, R., Grebely, J., Dore, G., J., Gray, R. T., Day CA, Kimber, J., Degenhardt, L. (2017). Estimating the number of people who inject drugs in Australia. BMC Public Health, 17, 757. DOI: 10.1186/s12889-017-4785

Point 3: Methods: Some narrative of the approach to participation may be helpful – it is noted that the response rate is included in the narrative of the results section as well as in table 1.  This may well be important to include in the discussion as those responding were using the SIF an average 3 times more than the people who did not respond.  The non-participants also appear less likely to have had a referral made in the previous 12 months. 

Response 3: Additional narrative (underlined) on the approach has now been provided in Materials and Methods, “Clients in Stage 3 were approached by a SIF staff member who informed them about the study and determined eligibility. People were eligible to participate if they had attended the service during the data collection period (October-November 2017), had attended the service on at least one other day, and had not already participated. Participation was voluntary. The questionnaire was administered by a SIF staff member in a private clinic room. Interviews took approximately 20 minutes to complete. Answers were confidential. In-line with established practice, participants were remunerated $20. Interviews were conducted across all opening hours and on 19 days during the data collection period.”

Additional information about the site has also been provided in this section, “The service operates seven days per week and is open from 930am-930pm on weekdays and from 930am-530pm on weekends. Frontline staff are comprised of registered nurses and health education officers.”

In line with reviewer comments we have revised (see underlined) the discussion, “There are several limitations to this study. The surveyed sample represented about one-third of all clients who attended the SIF during the study period. Participants were clients with longer histories of SIF use, a greater number of visits, and accordingly, more referrals which may might have reduced variability in the data and the significance of predictors in the logit models.”

Point 4: Results; Any reason why the data were not analysed by gender/sex? Even a mention that there were no differences if that is the case is important.  

Response 4: In a supplementary analyses of those who identified a current health and/or social issue, for all categories of health and social issues investigated there was no significant difference between the proportion of males and females indicating that they were experiencing the issue. We have mentioned this in the Results, “For each category, there was no significant difference between the proportion of males and females indicating that they were experiencing the issue.”

We note that in the analyses of factors associated with willingness to discuss housing issues, drug use and treatment, and take-home naloxone training with staff (Table 4), gender was included but the variable did not yield a factor loading of ≥0.25 and thus was not included in the separate stepwise logistic regression models.

We did not look further at gender because there was no significant difference in that characteristic between survey participants and non-participants.

Point 5: ‘On social benefits’ may be better described as ‘government income support’ rather than ‘welfare benefits’ as in the narrative.   

Response 5: We have revised text throughout accordingly, including Figure 1.

Point 6: For Table 2 the *** is missing for p<0.001.

Response 6: Missing ‘***’ now included.

Point 7: Section 3.3: is the nearby community health service Kirkteon rd Centre?  if so this may be better described as a primary health care service targeting their services to more marginalised communities including PWID.  This is also noted in table 2 ‘attendance at a local community health service’.

Response 7: The nearby service is the Kirketon Road Centre. Section 3.3 has been revised as suggested by reviewer, “About half (48%) of participants reported that for the first time since their initial visit to the Sydney SIF they had attended a nearby primary healthcare service which targeted services to more marginalised communities including PWID (Figure 1).”

Table 2 also revised accordingly.

Point 8: Table 3: it is not clear how the 80/176 who answered the TH naloxone question is reported.  May be worth considering a line in the tables that includes the 80/176 already done training.  You could also consider the same for the hepatitis c testing and treatment question as no doubt many people will have already done treatment or know their current negative status because of previous testing.

Response 8: The results in Table 3 have now been more clearly reported. Additional rows have been included as suggested by the reviewer. The description of results have been revised (see underlined) accordingly, “Participants were also asked if they were interested to receive support for six key health and social issues the Sydney SIF is well-positioned to influence (Table 3). About two-thirds (63%) of participants requested support in at least one of the six key areas. Overall, the most common request for support related to take-home naloxone training (48%). Just under half (44%) of participants who had not previously received training were interested in being trained. The next most common request for support was for hepatitis C testing and treatment (25%). Most participants had previously been tested for hepatitis C. Among the minority that had not been tested, 18% were interested in testing and treatment (Table 3). Support for mental health (12%) was the least requested (Table 3) despite the high self-report of current mental health issues (64%, Table 2).”

Point 9: Is there any data on the proportion of overdoses that took place at the SIF – of so it may also be worth including as it is noted in para 2 of the discussion that SIF management of overdose ‘may have contributed to individuals staying alive longer’. 

Response 9: We do not collect data on the number of overdoses that occur outside the Sydney SIF so it is not possible to determine the proportion of overdoses that occurred at the SIF. A recent review by Belackova et al. (2019) established that a main outcome of SIFs generally is overdose management and decreased mortality and it has now been included to support the statement ‘may have contributed to individuals staying alive longer’.

Point 10: Discussion: I wonder if it is worth framing this on the improvement and remained stable ‘large proportion’ rather than on the people for which their health and social situation had ‘deteriorated’. I understand from your discussion and conclusion that you do not want to add further to the stigma and barriers to access that already exist for many using the Sydney SIF.

Response 10: We have framed the Discussion as suggested, “This study found that, over time, clients’ engagement with treatment and a local low-threshold primary healthcare service increased, a direct or indirect outcome of referrals they received at the Sydney SIF. The health and social situation remained stable for a large proportion of clients and improved for others. Nevertheless, over time, the overall health and social situation among people who attended the Sydney SIF deteriorated, highlighting the dynamic and transitional nature of clients’ needs over the course of their SIF involvement. This is perhaps not surprising given clients’ ongoing (although at times intermittent) substance use which is a highly stigmatised and criminalised behaviour, and likely to impact on their health situation, hamper employment opportunities and increase contact with police [20].”

The Conclusion has been framed accordingly, “SIF clients connect with health and social services over time and the referrals they receive onsite likely contribute to this. Many SIF clients are open to health and social support at SIFs even though they present primarily to administer drugs in a safer setting. However, the overall health and social situation of SIF clients is likely to deteriorate over time, albeit not uniformly. Clients who are most likely to respond to support are those with greatest need. Nevertheless, engaging and referring clients from low threshold programs is just the first step in linking them to appropriate care. More attention is needed to develop improved practices at the high-threshold programs to which they are referred if these opportunities are to be harnessed.”

Point 11: The one thing in table two that stick out for me that wasn’t really discussed was the huge increase in the number of people who were ‘engaged in treatment’ – from 61% to 93%.  It may be that there is a different understanding of what treatment means for people over the years they have been using the service. This is an important finding and is worth highlighting more if it is possible to understand the nuances of ‘engaged in treatment’.  Partly this is because it is one of the criticisms of SIFs that they do not work hard enough to get people in to treatment – but alternatively it may be problematic in that almost all the people are ‘engaged in treatment’ but they are still using the SIF regularly so the treatment is not working.   The need to describe the benefits of access to low threshold ‘treatment’ may be required here. 

Response 11: The reviewer has raised a good point and we have included a discussion of these results in the Discussion, “The large increase over time in the number of people who were engaged in drug treatment (from 61% at initial visit to 93% at the time of the survey) challenges the misconception that SIFs encourage drug use and do not do enough to link people into treatment [23]. On the other hand, if people who are engaged in treatment are still using the SIF then is the treatment working? Critics of SIFs often point to a false dichotomy between treatment and harm reduction [24] and believe both cannot exist simultaneously. In reality, SIFs can do both {Aubin, 2010 #40}. Treatment does not have to be the primary goal and support is growing for targeted low-threshold healthcare and treatment which offers services without attempting to control drug use {Islam, 2010 #41}.”

Point 12: Para 4 (lines 225-231), How was ‘at risk’ defined? Why would heroin use necessarily be a risk? Nothing on concomitant use of benzos and or alcohol which may be considered ‘a risk’.  The ‘previous research’ mentioned mixes Canadian data and Australian data despite the differences in drug market settings – this may be sorted by some more detailed discussion of the diversity of SIFs suggested for the introduction.      

Response 12: ‘At risk’ was a term used to describe specific factors in the model. We have clarified terminology in this section of the Discussion, “Multivariate analysis revealed that individuals more open to support were those who were homeless, used heroin, had experienced overdose, and self-identified need. Not surprisingly, unstable accommodation and experience of overdose outside the SIF were strongly associated with willingness to discuss housing issues and take-home naloxone training, respectively.  Research from different international settings has consistently shown that more marginalized people use SIFs {Belackova, 2019 #34}. Despite different settings {Belackova, 2019 #34} frequent SIF attendance increased the likelihood of referral to health and social services {Kimber, 2008 #12;Salmon, 2017 #20} and engagement in drug treatment {Wood, 2006 #27; DeBeck, 2011 #28}. However, we found that such indicators of SIF service use were weaker determinants for requesting support than variables indicative of marginalization (i.e., homelessness, heroin use), self-identified needs and existing contact with relevant services.”

Recent research suggests that despite different settings the profile of SIF users and extent of referral and treatment uptake are remarkably consistent internationally {Belackova, 2019 #34}. See underlined text above.

Point 13: Para 6:  it may also be worth mentioning that the Richmond SIF now is based within a ‘community health service’ on the North Richmond public housing estate.  I did try and find a cite – perhaps Jauncey, M. 2017. If not us, Who? Drug and Alcohol Review 36(S1):5.  

Response 13: Now mentioned (see underlined text) in the third to last paragraph of the Discussion, “Other examples include co-locating SIFs at detoxification services (i.e., Vancouver Insite {Gaddis, 2017 #30}) and healthcare facilities {BCCS, 2018 #31}. The Melbourne SIF is now based within a community health service on a public housing estate.”

Point 14: Limitations: It may also be worth highlighting the potential for social desirability bias as people want to show the team that they really do want to ‘make adaptive changes’. Also, it may be that people who are not functioning as well (much much worse) than they were 10 years ago are now no longer able to even get to the SIF so these people have not been included in the data. 

Response 14: Additional limitations now considered, “The potential for social desirability response bias may have influenced results. Similarly, data may reflect the health and social situation of higher-functioning SIF clients as only people who attended the service were interviewed.”

Refences:

Point 15: Note that the cite 5 has now been published in DAR as well.

Response 15: Reference has now been amended.

Reviewer 2 Report

This study provided important information related to the unintended outcomes of persons using a SIF. The relationship between low threshold and high threshold services is clearly illustrated in the discussion section. 

As a reviewer outside of Australia, I found that I needed more information in the manuscript to understand the differences in the practices of an Australian SIF and other agencies. Since this journal is aimed at all disciplines, it is important to provide a clear explanation for various aspects of the study.

For example, it would be helpful to know what some of the questions that are asked on the initial intake to the SIF were and how the questions asked in the survey were different. This discrepancy was noted in the limitations section of the paper. This helps to identify if the methodology and findings were appropriate.

Why were participants interviewed during stage 3 of their visit to the SIF? They would have been under the influence of the injected drug and this has an impact on their response and may also impact their willingness to follow through with a concern. Either explain the rationale for interviewing at stage 3 or define at what point of stage 3 did the interview occur.

I am unclear as to why 500 were interviewed and 182 were surveyed. Were all persons interviewed (i.e. "Do you have any concerns that you want to address today?")  and only 182 agreed to partake in the survey that compared their initial intake information to their present information?

The 3 stars are missing from the legend in table 2

Line 194- Why would be Aboriginal and/or Torres Strait Islander origin be a factor for naloxone? This was noted to be a difference but was not addressed in the results section. this information may be useful to persons outside of Australia if it pertains to cultural influences of Indigenous Peoples.

Table 4 contained information that was collected but not adequately addressed in the narrative. 

The authors addressed important points to be expanded upon to competently care for the PWID. I found myself interested in wanting to know if individuals were asked about reasons why they did not utilize services (e.g. was this barrier related, stigma related, or just not interested). It is understood that the SIF is an area that addresses the content around safe/safer injecting practices but the authors made a connection between the increased use of services among persons that have utilized the SIF for long periods of time. This information is not new but ways to engage persons in care that they wish to receive may be informed by understanding why the person did or did not seek services. This speaks to a strength-based lens of care to adequately address the needs of this highly stigmatized and marginalized population.

Author Response

Response to Reviewer 2 comments

This study provided important information related to the unintended outcomes of persons using a SIF. The relationship between low threshold and high threshold services is clearly illustrated in the discussion section. 

Point 1: As a reviewer outside of Australia, I found that I needed more information in the manuscript to understand the differences in the practices of an Australian SIF and other agencies. Since this journal is aimed at all disciplines, it is important to provide a clear explanation for various aspects of the study.

Response 1: Additional international context has now been included in the Introduction, see response to Reviewer 1. We note that the last paragraph of the Introduction provides information about the harm reduction principles the Sydney SIF and most other SIFs operate on. The first paragraph of the Materials and Methods section provides a summary of the Sydney SIF clinical model.

Point 2: For example, it would be helpful to know what some of the questions that are asked on the initial intake to the SIF were and how the questions asked in the survey were different. This discrepancy was noted in the limitations section of the paper. This helps to identify if the methodology and findings were appropriate.

Response 2: We have now provided more information about the survey items (see underlined) in the Materials and Methods, “The detailed health and social information collected from clients at registration during their initial visit is not routinely updated. A 42-item questionnaire to update that information was developed which was based on the questions asked at registration. The interviewer-administered questionnaire assessed a range of indicators including: housing status (current  accommodation type), employment and source of income (work status, income source including welfare, crime and sex work), physical (hepatitis C status and testing) and mental health (seen a doctor, psychiatrist or counsellor for any mental health issues), incarceration history, drug use in the past month (mode of administration and frequency of injecting in the last month), overdose history, engagement in drug treatment (12 treatment modalities types, with an option for other), and health service utilisation of 15 local services spaning primary healthcare, social support, mental health, drug treatment, PWID peer organisations, hospital emergency departments, with an option of other in the last 12 months.

Also, we have expanded (see underlined) on the relevant section in the Discussion/limitations, “Also, several questions in the survey were phrased differently than the questionnaire used at initial visit, limiting the strength of some comparisons. For example, for some items there were differences in the assessment periods (e.g., past week versus past month) and response categories (e.g., frequency categories of drug use, and types of health services accessed) between questions asked at initial visit (which for some individuals were based on items developed more than 10 years ago) and questions asked at the time of the survey.

Point 3: Why were participants interviewed during stage 3 of their visit to the SIF? They would have been under the influence of the injected drug and this has an impact on their response and may also impact their willingness to follow through with a concern. Either explain the rationale for interviewing at stage 3 or define at what point of stage 3 did the interview occur.

Response 3: Clients’ primary reason for attending the SIF is to use drugs. It is not uncommon for people to present in various states of withdrawal (‘hanging out’), highly anxious, and with an urgent need to use drugs. It would be unrealistic (and unethical) to conduct research interviews prior to drug use in this setting. Instead, as we do for all research on-site, we approached clients after drug use in the aftercare area (Stage 3). While some clients may be intoxicated, many are simply ‘chilling out’, talking with other clients and staff, reading, doing crosswords, making phone calls or using the client computer. Informed consent was a requirement prior to participation. Using drugs does not necessarily preclude an individual from providing informed consent. Clients who were too intoxicated to provide this were not interviewed and were approached again later or another day. In Materials and Methods we have clarified this for the reader (see underlined), “All individuals gave their informed consent for inclusion before they participated in the study. Clients who were unable to provide informed consent due to their level of intoxication were approached again later or on another day. The study was conducted in accordance with the Declaration of Helsinki, and the protocol was reviewed by the Human Research Ethics Committee of the South Eastern Sydney Local Health District (17/207).”

Point 4: I am unclear as to why 500 were interviewed and 182 were surveyed. Were all persons interviewed (i.e. "Do you have any concerns that you want to address today?")  and only 182 agreed to partake in the survey that compared their initial intake information to their present information?

Response 4: Of the 500 clients who visited the SIF during the data collection period 182 were surveyed. We have clarified this for the reader (see underlined) when describing the sample in the Results section, “We interviewed 36% (n=182) of all clients (n=500) who visited the Sydney SIF during the study period.”

It was not possible to record the number of approaches made to clients.

Point 5: The 3 stars are missing from the legend in table 2.

Response 5: Addressed previously.

Point 6: Line 194- Why would Aboriginal and/or Torres Strait Islander origin be a factor for naloxone? This was noted to be a difference but was not addressed in the results section. This information may be useful to persons outside of Australia if it pertains to cultural influences of Indigenous Peoples.

Response 6: It is unclear why this is the case, however, it suggests that this marginalized population are willing to engage. We have now noted this in the Discussion, “It is unclear why people who are Aboriginal or Torres Strait Islander were more open to support but it suggests that this marginalized population are willing to engage and warrants further attention.”

Point 7: Table 4 contained information that was collected but not adequately addressed in the narrative. 

Response 7: In discussing the results reported in Table 4 we focused on those with clinical utility. Note the Discussion now includes a consideration of Aboriginality as a factor in seeking support (see above). We did not discuss buprenorphine, while it was a statistically significant factor the effect size was too small (AOR=0.068) to have clinical relevance.

Point 8: The authors addressed important points to be expanded upon to competently care for the PWID. I found myself interested in wanting to know if individuals were asked about reasons why they did not utilize services (e.g. was this barrier related, stigma related, or just not interested). It is understood that the SIF is an area that addresses the content around safe/safer injecting practices but the authors made a connection between the increased use of services among persons that have utilized the SIF for long periods of time. This information is not new but ways to engage persons in care that they wish to receive may be informed by understanding why the person did or did not seek services. This speaks to a strength-based lens of care to adequately address the needs of this highly stigmatized and marginalized population.

Response 8: The reviewer has flagged an interesting area and one which we are currently investigating in a qualitative study that is looking at the barriers that people experience in accessing healthcare and social services and how the Sydney SIF can optimize engagement with our clients and referrals to relevant services. This work is currently ongoing and, at this stage too preliminary to report on here.